# Determinants and barriers in early tuberculosis treatment in children at a primary health care facility in Kampala, Uganda; A mixed methods study

**Wani Muzeyi**[1]*, **Esther Babirekere**[2], **Dennis Kalibbala**[3], **Achilles Katamba**[3], **Joanita Nangendo**[3,4], **Fred C. Semitala**[4,5,6], **Mary Nyanzi**[7], **Victor Musiime**[1]

**1** Department of Pediatrics and Child Health, School of Medicine, College of Health Sciences, Makerere University, Kampala, Uganda, **2** Mwanamugimu Nutrition Unit, Directorate of Pediatrics and Child Care, Mulago National referral hospital, Kampala, Uganda, **3** Clinical Epidemiology Unit, School of Medicine, College of Health Sciences, Makerere University, Kampala, Uganda, **4** Department of Internal Medicine, School of Medicine, College of Health Sciences, Makerere University, Kampala, Uganda, **5** Infectious Diseases Research Collaboration, Kampala, Uganda, **6** Makerere University Joint AIDS Program, Kampala, Uganda, **7** Department of Paediatrics, Kawempe National Referral Hospital, Kampala, Uganda.

* wanixxl@gmail.com

## Abstract

### Background

Tuberculosis (TB) remains a leading cause of mortality worldwide, with childhood TB posing unique diagnostic challenges due to its pauci bacillary nature. The World Health Organization emphasizes that these diagnostic difficulties hinder early detection, contributing to delays in treatment initiation, disease progression, and increased morbidity and mortality. Addressing these challenges is critical to achieving the global goal of ending TB as a public health threat by 2030. This study aimed to determine the median time to TB treatment initiation and explore the factors influencing early treatment among children under 15 years at Kisenyi Health Center IV.

### Methods

We conducted a retrospective cohort mixed-methods study. Quantitative data were obtained through a retrospective review of medical records for 152 children under 15 years treated for TB at Kisenyi Health Center IV between February 1, 2021, and February 28, 2023. The median time to treatment initiation was estimated using Kaplan-Meier survival analysis, while Cox proportional hazards regression identified determinants of treatment initiation. Qualitative data were collected through key informant interviews with healthcare workers involved in childhood TB care. Thematic analysis, guided by the Capability, Opportunity, and Motivation Behavior (COM-B) model, was used to identify barriers to early TB treatment.

### Results

A total of 152 children were included in the study. The median time to TB treatment initiation was 39.5 days (IQR: 30,80.9). Pulmonary bacteriologically confirmed TB was the only

**Data availability statement:** All relevant data are within the manuscript and its Supporting Information files.

**Funding:** This research study was supported by the Fogarty international center and National institute on Mental Health of the National institute of Health under award number D43 TW010037.The content is solely the responsibility of the authors and does not necessarily represent the official views of the National institute of Health. The funder had no role in the study.

**Competing interests:** The authors have declared that no competing interests exist.

significant determinant of early treatment initiation (HR: 0.54, 95% CI: 0.31–0.88). Key barriers to timely TB treatment included caregivers' poor knowledge of childhood TB, referral of children under five to the national referral hospital, inadequate diagnostic equipment and supplies, loss of community follow-up contacts, and high patient volumes.

## Conclusion

Children with TB experience significant treatment delays, underscoring the urgent need for more accessible and rapid diagnostic tools to increase the proportion of bacteriologically confirmed cases and reduce treatment initiation time. Strengthening decentralized TB diagnostic capacity and enhancing caregiver awareness could improve early detection and treatment outcomes.

## Introduction

Globally, an estimated 1,200,000 children under 15 years developed tuberculosis (TB) in 2019, accounting12% of the global disease burden in 2019 [1]. In Uganda, statistical modeling estimates that 15–20% of the TB occur in children [2]. However, an epidemiological evaluation in Kampala reported that children contributed only 7.5% of all the reported TB cases with 54% of these occurring in children under 5 years [3]. Despite a lower incidence of TB in children compared to adults, mortality is disproportionately high with over 20% of the deaths occurring among children under 15 years [1].

Early diagnosis and treatment initiation (within 30 days of symptom onset) are critical in reducing mortality and preventing severe complications such as TB meningitis, disseminated TB, and post-TB lung disease [3–7].

Due to their immature immune system, children are at particularly higher risk of rapid disease progression [5]. The global annual TB report notes that only 30% of TB patients are initiated on treatment in the recommended 30 days from onset of symptom [1]. Various other studies conducted among adults with TB have reported median time to treatment initiation to range from 50 to 70 days from symptom onset [8–10]. There is limited data TB treatment initiation specifically among children.

Barriers to early TB treatment initiation among children include delayed diagnosis due to nonspecific symptoms, challenges in microbiological confirmation due to the pauci bacillary nature of childhood TB, health system inefficiencies and caregiver related factors [11,12]. Stigma, limited access to diagnostic services, and inadequate knowledge about TB have also been reported to be major barriers to early treatment initiation in low resource settings [13,14]

The Uganda National TB and Leprosy Program recognizes the importance of early TB treatment in children, as timely diagnosis and early treatment initiation can reduce mortality to near zero (Ministry of Health, 2022; Holmberg et al., 2019). However, significant gaps remain in understanding the determinants of early treatment initiation and the barriers that hinder timely care. Given these challenges and the anticipated shortfall in achieving the WHO's 2030 TB elimination target [1], this study aims to determine the median time to TB treatment initiation, as well as the determinants and barriers to early TB treatment among children under 15 years at Kisenyi Health Center IV.

## Materials and methods

### Study design and setting

This was a retrospective cohort mixed methods study conducted at Kisenyi Health Centre IV in Kampala, Uganda. Kisenyi Health Centre IV is a public primary healthcare facility. It is

located in Kampala's central division, serving a diverse urban population, including residents from TB hotspots such as informal settlements. It serves as a first point of care for many presumptive tuberculosis cases, providing free TB diagnosis and treatment. The facility diagnoses and manages TB using GeneXpert MTB/RIF, smear microscopy and clinical assessment. Patients are identified through passive case detection, contact tracing, and symptom-based screening. Those with presumptive TB undergo GeneXpert testing as the primary diagnostic test. The clinic runs Monday to Friday and receives an average of 80 children with presumptive TB per month. It is staffed by 8 health workers, including 1 medical officers, 3 clinical officers, and 3 nurses and 1 counsellor all trained in TB management and childhood TB diagnosis.

## Study population

**Inclusion.** Children under 15 years who were treated for TB at Kisenyi Health Centre IV between 1st February 2021–28th July 2023.
Exclusion
Patients with missing data on symptom onset or treatment initiation dates were excluded from the study.

## Sample size

The sample size was determined using the Cox proportional hazards model sample size formula. We assumed a 95% confidence level, 80% power, and an estimated hazard ratio of 1.27 for TB treatment delay when using microscopy compared to GeneXpert to diagnosis TB. Given an expected 31% treatment delay due to health system factors, we estimated that at least 52 cases of delayed TB treatment were required [15], resulting in a total sample size of 167 participants.

## Sampling procedure

We used consecutive sampling, extracting all eligible patient records from the TB register until the required sample size was achieved. Data extraction was conducted using Kobo Collect Toolbox between 7th and 21st July 2023.

## Study variables

The TB register at Kisenyi HC IV contains structured patient data, including: Patient demographics (age, sex, residence), date of symptom onset, date of TB diagnosis and treatment initiation, TB disease classification; pulmonary bacteriologically confirmed(PCB) or pulmonary clinical disease(PCD), TB diagnostic method used (GeneXpert, smear microscopy, clinical diagnosis), HIV status and treatment history, Nutritional status. The dependent variable was time(days) from date of symptom onset to date of treatment initiation

## Data management and analysis

The data was exported to STATA 17.0 for analysis. Categorical variables were summarized as frequencies and percentages while continuous variables were summarized as mean with standard deviation for normally distributed data and median with interquartile range for skewed data. Kaplan Meier survival curves were used to estimate Median time from symptom onset to TB treatment initiation. The relationship between time to TB treatment initiation and the covariates was analyzed using a Cox proportional hazards model. Variables were considered for multivariate analysis if having a p<0.2. Factors with a p≤0.05 were considered to be significant determinants of time to initiation of TB treatment.

### Qualitative interviews

Key informant interviews (KIIs) were conducted among health workers involved in TB care at Kisenyi HC IV to explore barriers to early TB treatment among children.

Health workers were purposively selected based on role in TB care and years of experience in the TB clinic (>1 year). A total of 9 key informants participated.

Interviews were conducted between 7th and 21st August 2023 at Kisenyi HC IV. Each interview lasted 30–45 minutes and was conducted in English by the PI who has experience in conducting KII among health workers. Interviews were audio-recorded and data collection continued until thematic saturation was reached.

Data was transcribed verbatim and analyzed using thematic analysis in Open Code software through the following steps: Familiarization: Reading transcripts and identifying patterns, Coding: Assigning labels to recurring concepts, Category Development: Grouping codes into broader categories, Theme Identification: Deriving key themes related to barriers to early TB treatment. The COM-B model (Capability, Opportunity, Motivation – Behavior) [16,17] was used as a theoretical framework to describe barriers to early TB treatment.

### Ethical approval and consent

Ethical approval was sought from the school of medicine research and ethics committee was granted #**Mak-SOMREC-2022–536** with a waiver of consent for extraction of quantitative data from the TB register at Kisenyi health center IV. Written informed consent was obtained from all key informant interviewees.

## Results

### Quantitative

A total of 215 participants met the eligibility criteria, of these 50 participants had no clear outcome variable and another 13 were missing key independent variables. The remaining 152 participants were included in the analysis (Fig 1).

**Participant characteristics.** Just over half (50.7%) of the participants were under 5 years of age with a median age of 4 years. Most (82.2%) participants were referrals from the community through contact tracing. The majority (76.3%) of participants had clinically diagnosed pulmonary TB. Chest x-rays and gene experts were done on 57.9% and 63.2% of participants respectively. Over 90% of the participants presented with a cough and only 7.2% were HIV positive (Table 1).

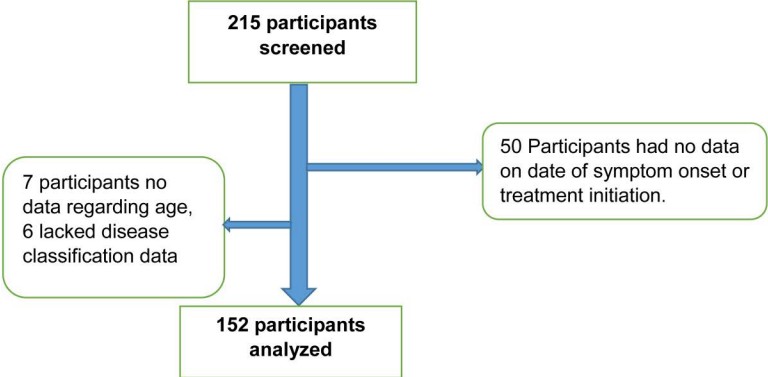

**Fig 1. Flow chart showing participant selection.**

**Table 1. Characteristics of 152 study participants.**

| Variable | Frequency(n=152) | Percent (%) |
|---|---|---|
| **Age (Years)** | | |
| <5 | 77 | 50.7 |
| ≥5 | 75 | 49.3 |
| **Referral type** | | |
| Community | 125 | 82.2 |
| Facility | 27 | 17.8 |
| **History of TB contact** | | |
| Yes | 76 | 50.0 |
| No | 76 | 50.0 |
| **Disease classification** | | |
| PBC | 36 | 23.7 |
| PCD | 116 | 76.3 |
| **History of cough** | | |
| No | 8 | 5.3 |
| Yes | 144 | 94.7 |
| **Chest X ray done** | | |
| No | 64 | 42.1 |
| Yes | 88 | 57.9 |
| **Gene expert done** | | |
| No | 56 | 36.8 |
| Yes | 96 | 63.2 |
| **HIV status** | | |
| Negative | 141 | 92.8 |
| Positive | 11 | 7.2 |
| **Nutritional status** | | |
| Malnutrition | 28 | 22.6 |
| Normal | 124 | 77.4 |

**Median time to TB treatment.** The median time to initiation of anti-TB medication among children at Kisenyi Health Center IV was 39.5 days with an IQR of (30,80.9) days.

**Determinants of time to TB treatment.** Only disease classification was a significant($p<0.05$) determinant of time to initiation of TB treatment. The results are summarized in Table 2 below.

## Qualitative

**Participant's characteristics.** A total of 9 key informant interviews (KII) were conducted among staff working at the TB clinic of Kisenyi Health Center IV. These include a medical doctor, 2 clinical officers, 2 registered nurses, and a laboratory technician, a counsellor, midwife and an enrolled nurse.

**Barriers of early TB treatment.** The COM-B Model of Behavior was used to describe barriers to early TB treatment. Five themes emerged from 3 COM-B domain. These are presented in Table 3 below.

**Poor knowledge of TB among caregivers.** Poor knowledge of TB among caregivers of children diagnosed with the disease led to unnecessary delays in treatment initiation. This emerged from 5 out of the 9 respondents interviewed, who highlighted that caregivers often lacked awareness of symptoms, appropriate diagnostic tests, and the urgency of early treatment.

**Table 2. Determinants of time to TB treatment.**

| Variable | Delayed treatment | | aHR | 95% CI | P value |
|---|---|---|---|---|---|
| | Yes(n=67) | No(n=85) | | | |
| **Age** | | | | | |
| <5 | 33(49.3) | 44(51.8) | 1.00 | | |
| ≥5 | 34(50.7) | 41(48.2) | 1.36 | 0.80-2.31 | 0.253 |
| **Referral type** | | | | | |
| Community | 57(85.1) | 68(80.0) | 1.00 | | |
| Facility | 10(14.9) | 17(20.0) | 0.75 | 0.34-1.65 | 0.469 |
| **History of TB contact** | | | | | |
| Yes | 30(44.8) | 46(54.1) | 1.00 | | |
| No | 37(55.2) | 39(45.9) | 1.3 | 0.77-2.10 | 0.362 |
| **Disease classification** | | | | | |
| PCB | 22(32.8) | 14(16.5) | 1.00 | | |
| PCD | 45(67.2) | 71(83.5) | 0.44 | 0.24-0.84 | **0.013** |
| **History of cough** | | | | | |
| No | 2(3.00) | 6(7.0) | 1.00 | | |
| Yes | 65(97.0) | 79(93.0) | 2.34 | 0.31-17.50 | 0.406 |
| **Chest X-ray done** | | | | | |
| No | 28(41.8) | 36(42.4) | 1.00 | | |
| Yes | 39(58.2) | 49(57.6) | 1.32 | 0.71-2.50 | 0.381 |
| **Gene expert done** | | | | | |
| No | 25(37.3) | 31(36.5) | 1.00 | | |
| Yes | 42(62.7) | 54(63.5) | 0.85 | 0.50-1.45 | 0.552 |
| **HIV status** | | | | | |
| Negative | 64(95.5) | 77(90.6) | 1.00 | | |
| Positive | 3(4.5) | 8(9.4) | 0.68 | 0.20-2.32 | 0.536 |
| **Malnutrition** | | | | | |
| No | 58(86.6) | 66(77.7) | 1.00 | | |
| Yes | 9(13.4) | 19(22.3) | 0.78 | 0.38-1.64 | 0.518 |

**Table 3. Application of the COM-B Model of Behavior used to describe barriers to early TB treatment.**

| COM-B component | Themes | Codes |
|---|---|---|
| **Psychological Capability** ((e.g., knowledge, memory of caregiver regarding TB in children) | • Poor knowledge of childhood TB among caretakers<br>• Referral of under 5 with presumptive TB to Mulago for Diagnosis | • Caretakers not sure whether the child has the symptoms and for how long<br>• Sometimes they refuse to do the CXR especially U5<br>• Referral for sputum induction and gastric aspirate<br>• We refer those that need Chest X ray |
| **Physical Opportunity** (How the health facility environment influences time to TB treatment) | • Inadequate equipment and supplies | • The reagents for gene expert sometime get out of stock<br>• GeneXpert machine breaks down |
| **Social Opportunity (How** inter-personal influences, social cues) | • Loss of community contacts | • They don't have transport to bring them for testing<br>• Their phone numbers are off, so we can't follow up<br>• They fear their children to be diagnosed with TB |
| | • High patient numbers | • We get up to 80 children with presumptive TB<br>• We are few staffs in the TB clinic |

A registered nurse emphasized the role of caregivers in facilitating early diagnosis:

*"Children won't give you history, it depends on the caretakers who at times don't know the children very well possibly because they don't stay with the child, don't know the duration of symptoms, have spent a lot of time treating the child with cough syrups"*, (KII2, Registered nurse).

Resistance to diagnostic tests was another major concern, as highlighted by a counselor

*"Some caregivers have refused to do certain tests like CXRs, gastric aspirate, lumbar punctures despite adequate counseling, they later come back when the child is too ill"*, (KII9, counselor).

**Referral of children under 5 with presumptive TB to the TB clinic at the national referral hospital.** Clinicians and laboratory personnel at the Kisenyi TB clinic identified significant challenges in diagnosing TB in children under five years. Four respondents (three clinicians and one laboratory technician) reported that due to the difficulty of obtaining samples from young children, referrals to Mulago National Referral Hospital were common

A laboratory technician described the referral process:

*"we only work on children where we can get samples, those that can't give us samples especially those under 5 years, we send them to Mulago for gastric aspirate and sputum induction"*, (KII6, Lab technician).

The referral system posed logistical and financial burdens for caregivers, as described by a medical officer:

*"TB is a disease of the poor. Many mothers lack transport to come to Kisenyi, yet we have to send them to Mulago for tests like chest X-rays. Imagine coming here for treatment, getting referred to Mulago for diagnosis, and then returning here for treatment. The queues at both facilities mean significant delays, sometimes stretching for weeks." (KII7, Medical Officer),"* (KII7, Medical officer).

## Inadequate equipment and supplies

A major barrier to timely TB diagnosis and treatment initiation was the lack of essential diagnostic tools and reagents. Six out of nine respondents acknowledged that the clinic was not well-equipped to handle pediatric TB cases effectively

A laboratory technician pointed out the recurrent shortages

*"Sometimes, we don't even have reagents to run GeneXpert or do sputum smears. This causes delays in diagnosis and treatment initiation"*, (KII6, Lab technician).

A clinical officer further elaborated on the diagnostic limitations:

*"Our capacity to investigate children for TB is significantly constrained. Tests like TB LAM, Mantoux, IGRA, and chest X-rays are unavailable here, so we refer patients, which prolongs diagnosis and delays treatment initiation"*, (KII3, Clinical officer).

## High patient numbers

Nearly all respondents cited the high volume of patients at the TB clinic as a major challenge, leading to delays in both investigation and treatment initiation.

A registered nurse highlighted the strain on resources:

*"We see about 200 patients with presumptive TB, of which 80 are children under 15 years, all these patients are handled by the same small team,"* (KII5, registered nurse).

*"In the laboratory, we receive over 80 samples per month for children under 15 years alone, when you add adult patients, investigations will inevitably delay,* (KII6, Lab technician).

## Loss of community contacts

Loss to follow up among children in contact with TB infected adults was another major concern, while village health teams(VHTs) trace child contacts, many caregivers failed to bring them for screening, leading to late stage diagnoses.

A counselor described the reasons given by caregivers as follows.

*"They (adults) don't bring their children for testing, claiming a lack of transport. Some also fear their children being stigmatized with a TB diagnosis,* (KII9, counselor).

*An enrolled nurse pointed out another challenge with follow up.*

*"The phone numbers they give us are sometimes incorrect. When we call to remind them to test their children or deliver results, the numbers are either off or wrong. Later these children return with advanced disease.",* (KII4, Enrolled nurse).

## Discussion

### Median time to TB treatment

The median time to TB treatment initiation among children at Kisenyi Health Center IV was high at 39.5 days from onset of symptoms, exceeding the WHO recommendation 30-day window within which treatment should be initiated. Only 67 (44.1%) of participants started treatment within this recommended period. Delays in treatment initiation increase the risk of disease progression, complications and ongoing transmission within households and communities. In Zimbabwe, 48% of the patients experienced patient delays beyond 30 days in seeking TB treatment services [15] and in in Australia only 31% of patients commenced treatment within 30 days with median time to initiation of treatment of 71 days however, although 86% of smear-positive cases received treatment within 3 days [7]. Similarly studies in Texas (52.56 days) [9] and France (67 days) [4] report longer median treatment times. while our study shows a shorter median time, it highlights delays in a primary health care facility catering to children in a low resource setting

### Determinants of time to TB treatment

The classification of TB significantly influenced treatment initiation timelines. Children with pulmonary clinical TB were 66% more likely to experience delays compared to those with pulmonary bacteriologically confirmed disease. A similar trend was reported in Britain where smear Positive TB cases were associated with a shorter time to treatment initiation [18]. This is because bacteriological confirmation prompts immediate TB treatment whereas clinically diagnosed disease requires additional investigations contributing to delays.

Efforts to develop and implement more sensitive and easily accessible diagnostic tools such as rapid molecular tests, are critical in reducing TB treatment delays at primary healthcare facilities.

## Barriers to early TB treatment

Using the COM-B model, we mapped barriers to TB treatment initiation across three behavioral domains: psychological capability, physical opportunity, and social opportunity [16,17].

**Psychological capability: Knowledge gaps.** A key barrier was poor caregiver knowledge of childhood TB, which delayed health-seeking behavior and treatment initiation. Similar findings have been reported, where limited caregiver awareness hindered TB contact tracing [19] and treatment adherence [20]. Caregiver-targeted health education sessions on childhood TB could improve early case detection and prompt treatment initiation.

Additionally, the referral of children under five years with presumptive TB to the national referral hospital emerged as another barrier, suggesting limited diagnostic capacity at Kisenyi Health Center IV. The need for gastric aspirates, sputum induction, and expert CXR interpretation led to delays due to logistical challenges and inadequate follow-up between health workers and caregivers. Strengthening on-site diagnostic capacity and enhancing TB staff training could minimize unnecessary referrals and treatment delays.

**Physical opportunity: Diagnostic and resource constraints.** Resource constraints were a significant barrier to timely TB treatment initiation. The facility faced frequent stock outs of TB diagnostics, including nutritional assessment tools (MUAC tapes, audiometers), limiting the screening and management of malnutrition, a known risk factor for TB. Additionally, the lack of an X-ray machine meant that children requiring imaging for TB diagnosis were referred to Mulago Hospital, resulting in substantial delays and increased caregiver burden. Equipping primary healthcare facilities with essential diagnostic tools would streamline TB diagnosis and treatment initiation.

**Social opportunity: Healthcare system barriers and community factors.** High patient volumes at Kisenyi Health Center IV overwhelmed existing TB clinic resources, leading to longer waiting times and delays in TB diagnosis and treatment initiation. Similar challenges have been reported in India [21] and Cambodia [13]., where limited human resources negatively impacted TB service delivery. Strengthening human resource capacity through increased staffing and task-sharing approaches—is crucial for reducing patient load and improving service efficiency.

A critical gap was the loss of childhood TB contacts at the community level. Although Kisenyi Health Center IV has established TB contact tracing mechanisms, caregivers often fail to bring exposed children for screening, citing transport barriers and stigma-related concerns. Similar findings in Cambodia highlight how poor community engagement and weak caregiver-health worker communication impede childhood TB case detection and management [13]. Incentivizing caregivers (e.g., transport reimbursements, community-based sample collection) could help improve childhood TB screening rates and early treatment initiation.

## Strength of the study

This study highlights challenges in infectious disease control in urban settings, the use of primary health system data provides valuable insights into real world barriers to timely pediatric TB treatment.

## Study limitations

Being retrospective study, we encountered missing data on key variables, which led to exclusions and this may have introduced selection bias. Additionally, the quantitative analysis was limited to variables available in the TB register, restricting the ability to explore other potential factors influencing treatment

## Conclusion

Children with TB at Kisenyi Health Center IV experienced a 10-day delay beyond the WHO-recommended treatment window, increasing their risk of disease progression.

Pulmonary bacteriologically confirmed TB was the only significant determinant of earlier treatment initiation, highlighting the need for more accessible and rapid TB diagnostic tools in children.

Key barriers to early treatment initiation included high patient volumes, inadequate diagnostic tools, frequent stock outs of TB diagnostics, referral delays for children under five, and poor caregiver knowledge of TB.

## Supporting information

**S1 File. mmed_data.**
(XLSX)

## Author contributions

**Conceptualization:** Wani Muzeyi, Fred C. Semitala, Victor Musiime.

**Data curation:** Wani Muzeyi, Dennis Kalibbala.

**Formal analysis:** Wani Muzeyi, Dennis Kalibbala.

**Funding acquisition:** Achilles Katamba, Fred C. Semitala.

**Investigation:** Wani Muzeyi, Victor Musiime.

**Methodology:** Wani Muzeyi.

**Resources:** Joanita Nangendo, Fred C. Semitala.

**Software:** Dennis Kalibbala.

**Supervision:** Esther Babirekere, Achilles Katamba, Fred C. Semitala, Victor Musiime.

**Validation:** Victor Musiime.

**Visualization:** Victor Musiime.

**Writing – original draft:** Wani Muzeyi, Mary Nyanzi, Victor Musiime.

**Writing – review & editing:** Wani Muzeyi, Achilles Katamba, Joanita Nangendo, Mary Nyanzi.

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
