## [Decision Letter · Decision Letter 0]

28 Jan 2025

PONE-D-24-37650DETERMINANTS AND BARRIERS IN EARLY TUBERCULOSIS TREATMENT IN CHILDREN AT A PRIMARY HEALTH CARE FACILITY IN KAMPALA, UGANDA; A MIXED METHODS STUDYPLOS ONE

Dear Dr. Muzeyi,

Thank you for submitting your manuscript to PLOS ONE. After careful consideration, we feel that it has merit but does not fully meet PLOS ONE’s publication criteria as it currently stands. Therefore, we invite you to submit a revised version of the manuscript that addresses the points raised during the review process.

**ACADEMIC EDITOR: **
**Please address all the issues raised by the reviewers. **

We look forward to receiving your revised manuscript.

Kind regards,

Novel N Chegou, Ph.D

Academic Editor

PLOS ONE

Journal Requirements:

 “This research study was supported by the Fogarty international center and National institute on

Mental Health of the National institute of Health under award number D43 TW010037.The content is solely the responsibility of the authors and does not necessarily represent the official views of the National institute of Health. The funder had no role in the study.”

“The authors declare no competing interests”

Reviewers' comments:

Reviewer's Responses to Questions

**Comments to the Author**

1. Is the manuscript technically sound, and do the data support the conclusions?

Reviewer #1: Partly

Reviewer #2: Partly

2. Has the statistical analysis been performed appropriately and rigorously? 

Reviewer #1: Yes

Reviewer #2: I Don't Know

3. Have the authors made all data underlying the findings in their manuscript fully available?

Reviewer #1: Yes

Reviewer #2: Yes

4. Is the manuscript presented in an intelligible fashion and written in standard English?

Reviewer #1: Yes

Reviewer #2: Yes

5. Review Comments to the Author

Reviewer #1: General overview

This is an important piece of work on a critical subject of early diagnosis and treatment of Pulmonary Tuberculosis in children less than 15 years. It is fairly well written but there are areas that need to be addressed to strengthen the manuscript.

Abstract

Capitalize the first letter when writing world health organization.

It is not clear how diagnosis of TB in children less than 15 years is a barrier to ending TB. Is it the delay, under-diagnosis, misdiagnosis, etc?

Methods: What was the study design? Mixed methods refer to methods of data collection (quantitative and qualitative) and not a study design.

The methods section needs to be reorganized to reflect the key aspects of design, study size, sample size, data collection and analysis. Currently, the authors present methods for quantitative data collection, then analysis and back to qualitative data collection.

Regarding the stated period of 1st-28th Feb 2023, it is not clear whether this refers to the time during which data was collected or that the participants were those that were initiated on TB treatment during this time. In the manuscript main text, the average number of patients with TB per month is indicated as 80.

Results IQR refers to Interquartile range and therefore should be presented as a range rather than a single figure.

How does referral of children less than 5 years to a national referral hospital become a barrier? It is delayed referral, delays in assessment at the national referral hospital or caregivers delaying going when referred?

The loss of community contacts as a barrier is also not clear.

Introduction

Paragraph 3: The 1st sentence is too long and the second part regarding post tuberculosis lung disease is unclear.

Reference 8 should be about a paper describing the natural history on TB as stated. However, this very reference is a review paper discussing highlights of the Global TB report 2020 and not a research article.

The manuscript is about time to initiation of TB treatment and related determinants and barriers. The introduction section should therefore focus on these 3 aspects. I note that there is no literature presented on determinants and barriers. The authors provide literature on delays to initiation of TB treatment with reference to studies among adults. Are there no studies in children which would be relevant to the study? If Yes, this needs to come out clearly. The factors at play in the different age groups are likely to be different.

Study design and setting

The ‘mixed methods’ indicates that the study had both quantitative and qualitative data collection methods, but it is not a study design. Clarify if this was a cross sectional, cohort, case-control or any other study design.

More details on the services. For example, what diagnostic tests for TB are available? To what extend are they accessible? What is the common practice for identifying and initiating TB treatment?

Give more information about the 8 HWs that run the clinic. Cadre, skills set etc.

What kind of patients attend the health centre given that it is in the city? Any referrals from other facilities to the TB clinic?

Regarding the number of patients with TB seen per month, this is a relatively high number (20 per week). It is imperative that you describe the type of patients attending this clinic, where they come from. For example, are they from a known TB hotspot?

Study population

This is blank. Describe the study population.

Add a sub-title for procedures or data collection. Then you can proceed to describe how the quantitative and qualitative data was collected.

The study methods for the quantitative part of the study are confusing. The first paragraph indicates that data was extracted from the registers and then later on, enrolment of study participants is mentioned.

Provide more information about the TB register. What kind of information is recorded and with what level of accuracy? For example, one of the study objectives was to determine the time from start of symptoms to initiation of treatment. Does the register capture such information?

The authors need to provide detailed and clear information on the methods including study design, setting, study population, eligibility criteria, sample size estimation and sampling process for both the qualitative and quantitative parts of the study.

The data collection procedures need to be well described, right from identifying potential participants/screening, consent and assent and tools used to collect the data, such as questionnaires, etc. I note that some of the data collection processes for the qualitative part are described under data management. This information should be transferred to the right section.

Data management and analysis

This section should be presented reflecting the objectives of the study and the corresponding method used to manage and analyse data.

The authors need to clearly describe how the data for the different objectives was analysed.

a) Median time to initiation of treatment

b) Determinants of initiation of treatment

c) Barriers to timely initiation of treatment

Ideally, when analysing any data set, one would first explore the data to check whether it is normally distributed or skewed and decide on whether to present as average (for normally distributed data) or median and IQR for skewed data. In this study, the 1st objective was very specific on median time to initiation of treatment. Was this based on the assumption that the data would be skewed?

Kaplan Mier survival curves were used to analyse the time to initiation of treatment. Please present the results of the graphs in the main text or as supplementary material for review and validation.

Results

The flow chart needs to be improved by providing details on the missing data. What is the outcome variable that was noted to be missing? Provide details on the independent variables that were missing and the corresponding number of participants in which they were missing.

Table 2 is missing.

Table 3 and 4 can be combined in one table

The results from the qualitative part of the study were based on what the health workers think and not necessary the truth and should be discussed as such.

Discussion

The terms time to initiation, median time, time interval are used interchangeably in the first paragraph but mean different things.

The authors provide information that compares the results with studies for other parts of the world. This is good. However, how does the results compare with the WHO recommendation and what are the implications?

Reviewer #2: The introduction is very short. The importance and necessity of the study should be fully explained.

The discussion to be completed in based on the relevant articles.

The findings to be completed in based on the purpose of the study.

The results of the study do not expand the boundaries of science.

Sources should be edited according to reference writing guidelines and journal format.

The strengths and limitations of the study to be mentioned.

It is suggested to remove the references before 2015 and replace them with new ones.

6. PLOS authors have the option to publish the peer review history of their article (what does this mean? ). If published, this will include your full peer review and any attached files.

**Do you want your identity to be public for this peer review?** For information about this choice, including consent withdrawal, please see our Privacy Policy .

Reviewer #1: No

Reviewer #2: No

---

## [Author Response · Author response to Decision Letter 1]

15 Feb 2025

The first letter of world health organization has been capitalized.

Childhood TB is pauci bacillary in nature. This makes it particularly difficult to diagnose, contributing to treatment delays. This has been made clear in the abstract.

The study design has been revised to retrospective cohort mixed methods study.

Sample size, data collection and analysis methods have all been revised.

The authors appreciate the reviewer’s insight on this subject.

The study included children under 15 years who were treated for TB at Kisenyi Health Centre IV between 1st February 2021 to 28th July 2023. Data was collected between 7th and 21st July 2023. This has been made more clear under study population and data collection.

Yes, IQR range is indeed a range and it has been revised as guided (30,80.9)

The referral system posed logistical and financial burdens to the caregivers of these children and this resulted into delays in treatment initiation.

With regards to loss of community contacts, caregivers would not bring childhood contacts of adults with TB to the hospital for early disease detection leading to late disease presentation.

These have been clarified under qualitative results

The introduction has been revised as guided. The first paragraph gives the global disease burden followed by local disease burden.

The said reference has been replaced with the actual WHO annual TB report. The author thanks the reviewers for this observation.

The introduction has been further refined to focus on time to initiation, determinants and barriers as guided.

The limitations in children’s literature on the subject matter have been clearly stated.

The study design has been revised to retrospective cohort mixed methods

The additional details with regards to services at the TB clinic have been added as advised under study design and setting.

More information is provided about the HW workers interviewed for the qualitative study under qualitative results

Kisenyi health center also serves informal settlements contributing to the high number of TB patients.This has been explained very well under study setting.

Study population, sampling procedure and sample size sub sections have been added as you have advised.

The collection of quantitative data was through data extraction from the TB register. This has been made clear under sampling procedure.

The TB register has been well described under a section of study variables. We again thank the reviewers for this insight.

Quantitative and qualitative sections have been re organized to make it easier on the reader.

The inclusion and exclusion has been clearly stated under the study populations.

The analysis plan has been revised as you have guided.

The analysis plan for the descriptive statistics has been added.

Kaplan Meir curves were used to estimate median time to treatment initiation and cox .proportional hazards regression was used for assessing determinants of time to initiation, Thematic analysis was used to summarize qualitative data.

This is all presented under data management and analysis.

More details have been added on the flow chart clarify exactly what variables had missing information

Tables have been merged as advised

The discussion has been revised as guided by the reviewers.

Recommendations on key research findings are within the text of the discussion.

The introduction has been expounded further to highlight importance and necessity of the study.

The limitations of the study have stated as advised

---

## [Decision Letter · Decision Letter 1]

11 Mar 2025

DETERMINANTS AND BARRIERS IN EARLY TUBERCULOSIS TREATMENT IN CHILDREN AT A PRIMARY HEALTH CARE FACILITY IN KAMPALA, UGANDA; A MIXED METHODS STUDY.

PONE-D-24-37650R1

Dear Dr. Muzeyi

We’re pleased to inform you that your manuscript has been judged scientifically suitable for publication and will be formally accepted for publication once it meets all outstanding technical requirements.

Kind regards,

Novel N. Chegou, Ph.D

Academic Editor

PLOS ONE

Additional Editor Comments (optional):

Reviewers' comments:

Reviewer's Responses to Questions

**Comments to the Author**

1. If the authors have adequately addressed your comments raised in a previous round of review and you feel that this manuscript is now acceptable for publication, you may indicate that here to bypass the “Comments to the Author” section, enter your conflict of interest statement in the “Confidential to Editor” section, and submit your "Accept" recommendation.

Reviewer #1: All comments have been addressed

Reviewer #2: (No Response)

2. Is the manuscript technically sound, and do the data support the conclusions?

Reviewer #1: Yes

Reviewer #2: (No Response)

3. Has the statistical analysis been performed appropriately and rigorously? 

Reviewer #1: Yes

Reviewer #2: (No Response)

4. Have the authors made all data underlying the findings in their manuscript fully available?

Reviewer #1: Yes

Reviewer #2: (No Response)

5. Is the manuscript presented in an intelligible fashion and written in standard English?

Reviewer #1: Yes

Reviewer #2: (No Response)

6. Review Comments to the Author

Reviewer #1: The manuscript has greatly improved and reads well.

A few points to consider

Address a few edits- references/citations- 2 styles used

Define the abbreviations (PBC and PCD) in table 1

Qualitative results: Under the sub-heading ‘Poor knowledge of TB among caregivers’, the quotes do not seem to point to lack of knowledge/awareness. This seems to be the interpretation of the authors. What these quotes indicate are issues like the caregiver not knowing the duration of the symptoms, probably because they did not stay with the child. This is not the same as lack of knowledge of the disease. Similarly, the refusal to do certain tests does not indicate lack of knowledge. It is important to note, these are views of the health workers and not the caregivers, and they should be presented as such. They may not be the actual barriers, but what the health workers think/perceive as barriers.

Reviewer #2: (No Response)

7. PLOS authors have the option to publish the peer review history of their article (what does this mean? ). If published, this will include your full peer review and any attached files.

**Do you want your identity to be public for this peer review?** For information about this choice, including consent withdrawal, please see our Privacy Policy .

Reviewer #1: No

Reviewer #2: No

---

## [Editor Report · Acceptance letter]

PONE-D-24-37650R1

PLOS ONE

Dear Dr. Muzeyi,

I'm pleased to inform you that your manuscript has been deemed suitable for publication in PLOS ONE. Congratulations! Your manuscript is now being handed over to our production team.

Kind regards,

on behalf of

Prof Novel Njweipi Chegou

Academic Editor

PLOS ONE